# PD-1-Associated Gene Expression Signature of Neoadjuvant Trastuzumab-Treated Tumors Correlates with Patient Survival in HER2-Positive Breast Cancer

**DOI:** 10.3390/cancers11101566

**Published:** 2019-10-15

**Authors:** William P. D. Hendricks, Natalia Briones, Rebecca F. Halperin, Salvatore Facista, Paul R. Heaton, Daruka Mahadevan, Suwon Kim

**Affiliations:** 1Integrated Cancer Genomics Division, Translational Genomics Research Institute, Phoenix, AZ 85004, USA; whendricks@tgen.org (W.P.D.H.); nbriones@tgen.org (N.B.); rhalperin@tgen.org (R.F.H.); sfacista@tgen.org (S.F.); 2Cancer and Cell Biology Division, Translational Genomics Research Institute, Phoenix, AZ 85004, USA; paulheaton@email.arizona.edu; 3Department of Basic Medical Sciences, University of Arizona College of Medicine-Phoenix, Phoenix, AZ 85004, USA; 4Department of Medicine, University of Arizona College of Medicien-Tucson, University of Arizona Cancer Center, Tucson, AZ 85719, USA; dmahadevan@uacc.arizona.edu

**Keywords:** breast cancer, HER2, trastuzumab, gene signature, PD-1, survival outcome, tumor evolution

## Abstract

The therapeutic HER2-targeting antibody trastuzumab has been shown to elicit tumor immune response in a subset of HER2-positive (HER2+) breast cancer. We performed genomic and immunohistochemical profiling of tumors from eight patients who have completed multiple rounds of neoadjuvant trastuzumabb to identify predictive biomarkers for trastuzumab-elicited tumor immune responses. Immunohistochemistry showed that all tumors had an activated tumor immune microenvironment positive for nuclear NF-κB/p65RelA, CD4, and CD8 T cell markers, but only four out of eight tumors were positive for the PD-1 immune checkpoint molecule, which is indicative of an exhausted immune environment. Exome sequencing showed no specific driver mutations correlating with PD-1 positivity. Hierarchical clustering of the RNA sequencing data revealed two distinct groups, of which Group 2 represented the PD-1 positive tumors. A gene expression signature that was derived from this clustering composed of 89 genes stratified HER2+ breast cancer patients in the TCGA dataset and it was named PD-1-Associated Gene Expression Signature in HER2+ Breast Cancer (PAGES-HBC). Patients with the Group 2 PAGES-HBC composition had significantly more favorable survival outcomes with mortality reduced by 83% (hazard ratio 0.17; 95% CI, 0.05 to 0.60; *p* = 0.011). Analysis of three longitudinal samples from a single patient showed that PAGES-HBC might be transiently induced by trastuzumab, independent of clonal tumor expansion over time. We conclude that PAGES-HBC could be further developed as a prognostic predictor of trastuzumab response in HER2+ breast cancer patients and be potentially used as an alternative biomarker for anti-PD-1 therapy trials.

## 1. Introduction

HER2-positive (HER2+) breast cancer constitutes 15–20% of breast tumors that express the HER2/ERBB2 receptor, associated with aggressive disease and poor patient outcome [1]. The adjuvant use of trastuzumab, a monoclonal antibody that targets the HER2 receptor, combined with chemotherapy, has significantly improved patient outcomes in HER2+ breast cancer with 46–56% reduction in recurrence rates and a 37% improvement in overall survival [2,3,4]. However, 20–30% of the patients do not benefit from trastuzumab and even those who initially benefit experience disease recurrence in 10 years, which indicates a need for the identification of the factors and biomarkers that determine therapy response in HER2+ breast cancer [4].

Genomic investigations have shown that HER2+ breast cancer is heterogeneous, but intrinsic molecular subtypes, such as HER2-enriched, HER2-luminal, and HER2-basal subtypes, can further classify it [5]. Trastuzumab was shown most effective in the HER2-enriched subtype with a 70% pathologic complete response (pCR) rate, as compared to 34–36% pCR rates in the HER2-luminal subtypes [6]. Nevertheless, these variable pCR rates display that intrinsic subtyping alone is not adequate to predict trastuzumab response.

Several gene expression signatures have been identified in HER2+ breast cancer associated with response to chemotherapy alone or combined with trastuzumab in clinical trials [6,7,8,9]. More recently, an integrative analysis that was based on a predictive algorithm combining multiplatform information on HER2+ tumors and previously published gene expression signatures concluded that multiple factors are predictive of therapy response, including *TP53* mutations, HER2-enriched intrinsic molecular subtype classification, and an immune signature [10]. In addition, exome sequencing of tumors pre- and post-neoadjuvant chemotherapy and trastuzumab showed the presence of therapy-resistant subclones in pre-treatment tumor samples, which indicated that intra-tumor heterogeneity also plays a significant role in therapy resistance [11]. While these studies attest to the complexity and heterogeneity of HER2+ breast cancer in therapy response, biomarkers that are specific to trastuzumab response have not been well-defined. 

Trastuzumab’s anti-cancer mechanism involves not only inhibitory binding to the HER2 receptor, but also eliciting antibody-dependent cellular cytotoxicity via activating tumor-killing immune cells [12]. Concordantly, tumor-infiltrating lymphocytes (TILs) have been associated with improved patient survival in an adjuvant trastuzumab-based clinical trial [13]. However, a recent study showed that a TIL-associated gene signature correlated with a response to chemotherapy but not with trastuzumab response in HER2+ breast cancer [14]. Thus, the contribution of TILs in trastuzumab-specific response has remained unclear. 

Programmed cell death protein 1 (PD-1) is a member of the CD28 cytotoxic T lymphocyte antigen-4 (CTLA-4) family known as immune checkpoint molecules [15]. The function of immune checkpoint molecules is to limit immune activity in the course of normal immune response, without which immune-mediated cellular toxicity can be detrimental [15,16]. The expression of PD-1 or its ligand PD-L1 in the tumors is thought to indicate a suppressive and/or exhausted immune environment that allows tumor evasion of immune-mediated killing [17]. Corroborating this idea, immune checkpoint inhibitors including anti-PD-1 agents have emerged as successful anti-cancer therapies for a subset of patients, which indicates that a PD-1 positive immune environment contributes to cancer [18]. In breast cancer, the presence of PD-1 positive TILs was associated with aggressive tumors, HER2+ status, and poor patient survival in breast cancer [19,20]. More recently, an elevated expression of PD-L1 was associated with aggressive tumor features, but nonetheless correlated with better response to chemotherapy, which suggested that an exhausted immune environment might play a role in therapy response [14,21,22].

A clinical trial showed that tumors that were exposed to single-dose trastuzumab had an increased immune presence, including PD-1 positive TILs in 17–40% HER2 tumors, which suggested that trastuzumab might directly elicit specific tumor immune responses in a subset of tumors [23]. However, this immune signature could not predict therapy response at baseline and thus had limited utility as a predictive biomarker [23]. We hypothesized that genomic footprints associated with trastuzumab-induced immune response were identifiable in post-treatment tumors, which could be used to evaluate primary tumors to predict therapy response at diagnosis. To this end, we analyzed eight HER2-positive breast tumor samples from patients who had completed two or more cycles of neoadjuvant trastuzumab and evaluated the genomic features of these tumors with a focus on the PD-1 positive tumor immune environment.

## 2. Results

### 2.1. Variable PD-1 Expression in HER2-Positive Tumors from Patients Treated with Neoadjuvant Trastuzumab

Eight FFPE tumor samples were evaluated from patients who had been diagnosed with HER2+ breast cancer and treated with two or more cycles of neoadjuvant trastuzumab prior to definitive surgery in order to assess the immune environment of HER2+ breast tumors post-trastuzumab treatment. As all samples were from residual disease post-trastuzumab treatment, these cases were considered to be trastuzumab-resistant. Patients were between 36 and 61 years of age. Four patients were reported lymph node-positive and three were lymph node-negative for tumor cells, while the lymph node status of one patient was not provided. Tumors that were obtained by surgery were 0.2–4.5 cm and grade II–III. Three tumors were reported to express estrogen receptor (ER) at a varying degree 40–90%, while five tumors were reported not to express ER. Table 1 summarizes the patient information obtained through clinical reports, including TNM staging.

IHC staining of the tumor samples was performed for immune-related markers, including NF-κB/p65RelA, CD8, CD4, and PD-1. The results showed that all seven tested tumors were positive for high levels of nuclear NF-κB/p65RelA, a key molecule that mediates the transcription of inflammatory cytokines, as previously reported in HER2+ breast tumors [24]. All seven tumors were also positive for CD8+ T cells, and six out of seven were positive for CD4+ T cells, indicating the presence of T cell populations in HER2+ tumors that were treated with neoadjuvant trastuzumab. The staining of NF-κB/p65RelA, CD8, or CD4 could not be determined for one sample, S6, due to the limited tissue availability. Despite the presence of T cells in all tumors tested, PD-1 expression was detected in only four out of eight tumors (S4, S5, S10, and S11, Table 2), constituting 50% of this tumor cohort. Figure 1 shows IHC examples for PD-1 negative and PD-1 positive tumors (IHC summary in Table 2). These results demonstrated that T cells were present in all HER2+ tumors that were treated with trastuzumab, but only a subset of tumors had a PD-1 positive immune environment. Any statistical association between PD-1 and clinicopathologic parameters could not be determined due to the small cohort size. Nonetheless, one out of four LN+ tumors was positive for PD-1, while one out of three ER+ tumors was PD-1 positive, which suggested that neither factor plays a major role in a PD-1 positive tumor environment.

### 2.2. No Putative Tumor Driver Mutations Associated with the PD-1 Positive Tumor Immune Environment

To determine whether specific genetic alterations were associated with the PD-1 positive immune environment in post-trastuzumab tumors, we carried out DNA sequencing of tumor material from FFPE sections for these cases. We searched for somatic, putatively pathogenic single nucleotide variants (SNVs) and copy number variants (CNVs), as summarized in Table 2. Candidate somatic variants were identified by first selecting SNVs in coding and splice regions, selecting CNVs with two-copy gain or loss. We then filtered for mutations impacting COSMIC cancer genes (COSMIC v82) and those that have not been described in the general population (maximum population allele frequency from ExAC and 1000 Genomes ≤ 0.01). The total number of rare SNV/CNVs impacting cancer genes ranged from three to 25 per tumor. We found that two out of eight tumors harbored inactivating mutations in *TP53*, while two tumors contained activating mutations in *PIK3CA*, C420R, and Q546R, both of which have been reported in breast cancer [25]. Of note, two out of four PD-1 positive tumors bore a *TP53* mutation, while one out of four bore a mutation in *PIK3CA,* which suggested that the mutations in these two cancer driver genes are not associated with PD-1 positivity. Other notable mutated cancer genes were *BRCA2*, *NF1*, and *GATA3*, each occurring once in non-overlapping tumors (Appendix A). The mutation frequencies for these genes approximated those that were reported previously in breast cancer, which supported the relevance of the tumor cohort in this study [26]. The complete list of filtered SNVs and CNVs in each tumor is shown in Appendix A. No specific alterations in cancer genes were associated with PD-1 positive tumors, suggesting that the PD-1 positive immune microenvironment in trastuzumab-treated HER2+ tumors might not be correlated with specific gene mutations, at least not to one single recurrent mutation.

### 2.3. Two Distinct Tumor Groups Identified by Gene Expression Analysis

We next examined whether a gene expression profile(s) was associated with the PD-1 positive tumor immune environment by sequencing RNA extracted from the FFPE sample sections. We utilized HTSeqCount values that were generated from STAR aligner’s RNA-seq output to perform unsupervised clustering of the normalized expression levels of the 477 significantly differentially expressed genes with a Log2FC ≥ 2 or Log2FC ≤ –2 from DESeq2 comparison of tumors to the commercially available Universal breast control (see Methods). The clustering analysis separated tumors into two distinct groups (Figure 2A). Posterior DESeq2 analysis of Group 1 vs. Group 2 resulted in 89 genes in common between the two differential gene expression analyses (Figure 2B, the 89 genes listed in Appendix A). Supervised clustering with HTSeqCounts from the STAR aligner of the RNA-Seq data using the 89 genes also resulted in the clear separation of two groups with an associated *p*-value = 0.00036 from Fisher’s test. Group 1 was defined as tumors with elevated expression of 25 genes (top 24 genes in Figure 2C) with Log2FC values ≥ 2 and the reduced expression of 64 genes (bottom 64 genes in Figure 2C) with Log2FC values ≤ –2 from DESeq2 analysis. Conversely, Group 2 was defined as tumors with a reduction of the 25 and elevation of the 64 genes. Three tumors—S2, S6, and S7—constituted Group 1, while five tumors—S3, S4, S5, S10, and S11—Group 2. These groups also correlated with PD-1 status with Group 1 being PD-1-negative and all but one tumor in Group 2 being PD-1-positive. As with PD-1 positivity, no correlations were seen between the expression profiles and driver gene status.

### 2.4. Elevated Expression of Immune and Inflammatory Genes in Group 2

Pathway analysis with DAVID v6.8 [27] of these 89 genes revealed several genes that were mapped to the cytokine-receptor and immune-related pathways, including *CXCL10*, *CXCL9*, and *CTLA-4* (see Appendix A) with a *p*-value of 0.018. CTLA-4 belongs to the functional family group of the immune checkpoint molecules that include PD-1 [15]. Cxcl10 and Cxcl9 are chemokines that primarily function as chemoattractants for T cells [28]. These genes were significantly more highly expressed in Group 2 tumors, which suggested that neoadjuvant trastuzumab might have elicited a robust T cell response in a subset of tumors, leading to a suppressive and/or exhausted immune environment. Consistent with this idea, four out of five tumors in Group 2 were positive for PD-1 by IHC (Table 2). Conversely, the Group 1 tumors showed reduced expression of the cytokine pathway genes and were negative for PD-1 IHC (Table 2). *PDCD1*, the gene encoding PD-1, or *CD274* encoding PD-L1, the PD-1 binding partner, did not show significant expression differences between Group 1 and Group 2 tumors (*PDCD1*, *p* = 0.064; *CD274*, *p* = 928; Figure 3A). A trend toward a higher *PCDC1* expression in Group 2 was observed, but by less than two-fold (average HTSeq counts of 99 in Group 1 versus 159 in Group 2; Figure 3A), which suggests that the expression level of PD-1 or PD-L1 might not provide an optimal indicator for a PD-1 positive tumor microenvironment (see below). Neither was *CXCL10* or *CXCL9* differentially expressed between groups (*CXCL10*, *p* = 0.058; *CXCL9*, *p* = 0.077; Figure 3B). However, the average expression of *CXCL10* and *CXCL9* was higher in Group 2 tumors by 22.8-fold and 12.3-fold, respectively (Figure 3B), which supports the idea that Group 2 tumors were “immune-active”, which may have resulted in a PD-1 positive tumor immune environment. Comparatively, the genes highly expressed in Group 1 tumors included *TAT,* which encodes a tyrosine aminotransferase and *VIT* that encodes vitrin, an extracellular matrix protein. *TAT* and *VIT* were expressed at higher levels with statistical significance in Group 1 tumors by 56.9-fold and 8.7-fold, respectively (*TAT*, *p* < 0.0001; *VIT*, *p* = 0.007; Figure 3C). Whether *TAT* or *VIT* are functionally related to the configuring of a tumor immune microenvironment is currently unknown (see Discussion).

### 2.5. The 89-gene Signature Correlated with HER2+ Breast Cancer Patient Survival

The 89-gene signature was used to stratify breast cancer patients in The Cancer Gene Atlas (TCGA) dataset in order to assess a clinical relevance of the gene signature identified in HER2+ tumors treated with neoadjuvant trastuzumab. Of the total cohort, 250 out of 1090 (23%) breast cancer patients contained the Group 1 signature, while 309 (28%) contained the Group 2 signature (Table 3). The HER2+ subset of TCGA breast cancers for this study was composed of 181 tumors scored “positive” in the “HER2final” category. Of the HER2+ subset, Group 1 patients constituted 20% (36 out of 181), while Group 2 constituted 29% (53 out of 181, Table 3). Other subtypes, including estrogen receptor-positive HER2-negative (ER+HER2−) and triple-negative breast cancer (TNBC), showed 20–22% Group 1 and 27–32% Group 2 patient distributions (Table 3). These distributions indicated that the 89-gene signature was not specific to HER2+ breast cancer or to any other subtype.

However, Kaplan-Meier analysis showed that the gene signature significantly stratified HER2+ breast cancer patients according to overall survival (Figure 4C vs. A,B,D). Patients with the Group 2 gene signature had improved survival by 83% when compared to the patient with the Group 1 gene signature (HR 0.17, 95% CI 0.05–0.60, *p* = 0.011 Figure 4C). The gene signature did not significantly stratify patients in other subtypes of breast cancer (all breast cancers, *p* = 0.061, Figure 4A; ER+HER2− breast cancers, *p* = 0.969, Figure 4B; TNBC, *p* = 0.101, Figure 4D). These results supported that the 89-gene signature was specific to HER2+ breast cancer. Moreover, as tumors in the TCGA dataset are pre-treatment primary tumors, these results indicated that the 89-gene signature might represent an intrinsic tumor composition that confers trastuzumab response in HER2+ breast cancer. We thus named the 89-gene signature, PD-1 Associated Gene Expression Signature in HER2+ Breast Cancer, PD-1-Associated Gene Expression Signature in HER2+ Breast Cancer (PAGES-HBC).

Of note, the HER2+ subset of TCGA breast cancers consisted of 139 estrogen receptor-positive (ER+) and 42 estrogen receptor-negative (ER−) tumors. While PAGES-HBC’s stratification of HER2+ER+ breast cancer patients was comparable to that of all HER2+ patients (HR 0.17, 95% CI 0.04–0.65, *p* = 0.012), we could not determine a significance in HER2+ER− patients, which was likely due to the small cohort size, including only 14 Group 1 and 6 Group 2 patients.

It is notable that the *PDCD1* (the PD-1 gene) or *CD274* (the PD-L1 gene) expression levels were not different between PAGES-HBC Group 1 and Group 2 in the HER2+ breast cancer TCGA cohort (*PDCD1*, *p* = 0.193; *CD274*, *p* = 0.637);which indicated that PAGES-HBC might serve as a better prognostic marker representing the PD-1 positive immune environment related to therapy response and patient survival in HER2+ breast cancer.

### 2.6. Transient Trastuzumab-Elicited Immune Activation Displayed in Longitudinal Samples from a Single Patient

We performed integrated analysis of three longitudinal samples from a single patient to further assess the immune environment and expression changes over time in the setting of trastuzumab-treated HER2+ breast cancer, including: needle biopsy at the time of diagnosis (primary tumor, H & E staining shown in Figure 5A,B), resected tumor at definitive surgery following four cycles of trastuzumab (S10 tumor listed in Table 1 and Table 2, H & E staining shown in Figure 5G,H), and chest wall biopsy at recurrence seven months post therapy (H & E staining shown in Figure 5M,N). IHC staining for NF-κB, CD4, CD8, and PD-1 showed that the primary tumor contained a mixed population of NF-κB/p65RelA positive and negative cells, the presence of CD4+ and a low number of CD8+ cells, and no staining for PD-1 (Figure 5C–F). The post-trastuzumab tumor showed high immune activation with distinct nuclear staining of NF-κB/p65RelA in the majority of tumor cells alongside a higher number of CD4+ and CD8+ cells, and numerous immune cells expressing PD-1 (Figure 5I–L). In contrast, the recurrent tumor was NF-κB/p65RelA, but relatively immune-quiet with no detectable CD4, CD8 or PD-1 positive cells (Figure 5O–R).

Gene expression analysis of the three samples supported the IHC results, revealing that, while PAGES-HBC in the primary tumor was significantly different from that in the post-trastuzumab tumor (*p* = 0.03), the primary and recurrent tumors showed no significant difference in PAGES-HBC (*p* = 0.65, Figure 6A). These results suggested that trastuzumab might have transiently induced gene expression changes. Figure 6B and C illustrate examples of PAGES-HBC genes that showed transient expression changes in the post-trastuzumab tumor. We also evaluated changes in the expression of *PDCD1* (PD-1) and *CD274* (PD-L1) in the longitudinal samples and found that both genes were highly expressed in the post-trastuzumab tumor, but relatively low in the primary and metastatic recurrent tumors (Figure 6D). Collectively, these results support the hypothesis that trastuzumab-elicited tumor immune activation might be transient in some patients, as was the case with our sample, but it needs to be surveyed in others.

### 2.7. Clonal Expansion of Single Nucleotide Variants (SNVs) in the Longitudinal Samples from a Single Patient

We next evaluated the longitudinal changes in putative driver mutations occurring in this single patient in order to look for genetic features that might correlate with changes in the tumor immune environment. SNVs that were detected in each sample were grouped by their pattern of abundance across the samples, which resulted in four “clonal variant groups”: The clonal variant group A was present in a high proportion of the tumor in all three samples, thus likely representing founder mutations that were not affected by trastuzumab (blue, Figure 7A). Concordantly, SNVs in the clonal variant group A included a *TP53* 949delC/Q317fs, likely a driver mutation shown to occur in many cancer types, including breast cancer [25,29]. The clonal variant groups B and C represented sub-clones that increased in abundance over time, which suggested that tumor cells containing these SNVs were therapy-resistant (red and green, Figure 7A). In contrast, the clonal variant group D was present in the primary tumor but mostly not detectable at the later time-points, suggesting tumor cells containing these SNVs were therapy-sensitive (orange, Figure 7A). The average proportion of cells in each sample expected to have the mutations in each clonal variant group is graphed in Figure 7B to visualize the extent of the clonal group evolution. Appendix A lists the SNVs in the clonal variant groups.

Pathway analyses with DAVID v6.8 [27] of each four variant clonal groupings with variants that had PASS and low quality control metrics did not reveal any specific functional relationships between the mutant genes in each group. The clonal group B and C genes that may potentially play a role in trastuzumab resistance numbered 77 and 161, respectively, rendering it difficult to distinguish between driver and bystander mutations (Appendix A). However, it was evident that, since the clonal variant groups B and C expanded over time through trastuzumab, these SNVs are unlikely to be responsible for the PD-1 positive immune environment present only in the post-trastuzumab sample (Figure 5 and Figure 6).

## 3. Discussion

We demonstrated that a subset of HER2+ breast tumors that were treated with neoadjuvant trastuzumab had a PD-1 positive immune environment. The gene signature that was derived from post-trastuzumab tumors, named PAGES-HBC, effectively stratified HER2+ breast cancer patients in the TCGA dataset. Specifically, patients with the Group 2 PAGES-HBC had significantly improved survival outcomes by 5.8-fold (log-rank *p* = 0.011, Figure 4C). As the TCGA dataset is composed of primary tumors, these results indicated that the genomic footprint of PD-1-associated trastuzumab response is already present in pre-treatment tumors. Thus, these results suggest that PAGES-HBC could be used to predict trastuzumab response in HER2+ breast cancer.

When considering the current paradigm that PD-1/PD-L1 allows the immune evasion of aggressive tumors associated with poor prognosis, our findings that the PD-1 positive PAGES-HBC is a good prognostic indicator in HER2+ breast cancer are counterintuitive. One potential explanation could be that PD-1 expression in post-trastuzumab tumors indicates tumor response to therapy, not a suppressive or exhausted immune environment associated with aggressive tumors. Consistent with this idea, the induction of PD-1 and other immune markers has been reported in a subset of tumors briefly exposed to trastuzumab, but were not prognostic at baseline [23]. In addition, our results of the longitudinal samples showed the increased expression of *PDCD1* and *CD274* only in the post-trastuzumab tumor (Figure 6D), supporting the notion that PD-1 expression is an indicator of trastuzumab response. In our study, tumors positive for PD-1 by IHC had nominally increased the expression of the *PDCD1* gene (<2-fold, *p* = 0.064, Figure 3A), which rendered the *PDCD1* gene expression level a suboptimal biomarker. PAGES-HBC could serve as a better predictor of trastuzumab response related to PD-1 induction, given the transient and variable induction of PD-1 in post-trastuzumab tumors. 

The blockade that was directed to PD-1 and its ligand PD-L1 has shown a durable long-term response in a subset of the patients with cancers including melanoma, non-small cell lung cancer, and kidney cancer [30]. Clinical trials of PD-1/PD-L1 blocking agents are ongoing in TNBC with promising 19–20% response rates [31,32]. In addition, a recent phase 1b-2 clinical trial of anti-PD-1 agent combined with trastuzumab in HER2+ breast cancer patients showed objective responses in patients with tumors positive for PD-L1 [33]. While PD-L1 IHC has been approved by the U.S. Food and Drug Administration as a companion biomarker for PD-1/PD-L1 blocking agents, variable staining cutoffs and high rates of false negatives indicated a needed improvement in a reliable biomarker for immune checkpoint blockade responders [34,35]. When considering that PAGES-HBC was prognostic despite the unknown trastuzumab treatment status in the TCGA dataset, PAGES-HBC might be predictive of therapy response in general, related to tumor immune response. This idea might be supported by the fact that PAGES-HBC itself was not specific to HER2+ breast cancer. In this case, PAGES-HBC could be tested for its utility as a biomarker not only for trastuzumab, but also for other immune-activating agents, including immune checkpoint blockade agents. Moreover, our study showed that PAGES-HBC preludes PD-1 expression in post-trastuzumab tumors in HER2+ breast cancer. These suggest that patients with PAGES-HBC may benefit from combination therapy of trastuzumab and an anti-PD-1 agent. A clinical trial using PAGES-HBC as a response indicator might identify such a patient cohort.

It is notable that PAGES-HBC was specifically prognostic in HER2+ breast cancer, which suggested a functional relationship between the PAGES-HBC genes and HER2 biology. The HER2 receptor has been shown to activate NF-κB directly and indirectly in breast cancer, which in turn NF-κB activates the transcription of numerous inflammatory genes [36,37,38]. PAGES-HBC includes several NF-κB target genes, including *CXCL10*, *CXCL9*, *CD80*, and *SELE* (Appendix A). These genes are also related to T cell functions. Thus, PAGES-HBC might represent a HER2/NF-κB/T cell signaling axis that determines therapy response in HER2 breast cancer. Functional investigation of the interplay between these signaling pathways may reveal the intrinsic and/or extrinsic factors that are responsible for PAGES-HBC.

*TAT* and *VIT* were among the group of genes in PAGES-HBC differentially expressed, but with limited information on the association with breast cancer. *TAT* encodes a liver enzyme tyrosine aminotransferase of which expression is induced by steroid hormones [39]. It has recently been implicated in chicken oviduct proliferation and tumorigenesis under the regulation of estrogen, which suggests its potential role in some cancers [40]. However, in the setting of PAGES-HBC, it is also possible that the enzymatic activity of TAT contributes to immune cell functions in the tumor microenvironment, as other metabolic enzymes, such as indoleamine 2,3-deoxygenases (IDO), have been implicated in immune cell regulation [41]. *VIT* encodes vitrin, an extracellular matrix protein that was initially discovered in bovine vitreous and more recently implicated in post-natal cartilage development [42,43]. While it is conceivable that vitrin-containing extracellular matrix might inhibit tumor immune response, this needs to be directly tested. Inasmuch as it is plausible that *TAT* and *VIT* may have direct functions in modulating the tumor immune microenvironment, it is also formally possible that these genes are under the same transcriptional regulation as the other genes in PAGES-HBC.

In the longitudinal sample analysis, we showed that PD-1 associated gene expression was transient and no longer detected in recurrent tumors (Figure 6). These findings clearly need to be validated in more than one patient sample, but highlight the importance of longitudinal samples in deciphering temporal dynamics of therapy response and resistance. We reported the clonal expansion of SNVs over time in longitudinal patient samples, suggesting that specific SNVs may provide selective advantage under trastuzumab treatment pressure (Figure 7). The results also demonstrated that therapy-resistant subclonal population existed in pre-treatment tumor, as has been reported previously [11]. These underscore the need to address intra-tumor heterogeneity in primary tumors. Further investigation of SNVs for their functional roles in therapy resistance might enable us to identify biomarkers and tractable targets to treat trastuzumab-resistance in HER2+ breast cancer at diagnosis.

## 4. Materials and Methods

### 4.1. Patient Samples and Tumor Tissue Microarray (TMA)

Fifteen formalin-fixed paraffin-embedded (FFPE) HER2-positive breast cancer patient samples consisting of 10 resected tumors and five biopsy samples collected between 2006 and 2011 were accrued from the University of Arizona Cancer Center (UACC) tissue bank according to the approved Institutional Review Board protocol # 0600000609R006 and through a Banner University Medical Center pathology honest broker agreement. A tumor tissue microarray (TMA) was constructed by extracting 1.0 mm diameter tissue punches from each tumor tissue block and embedding tissue punches into a fresh paraffin block that was manually completed with protocol H.17.2 from Tissue Acquisition Cellular Molecular Analysis Shared Resource (TACMASR) at the UACC. The TMA contained 31 tissue punches, including double punches from 15 independent tumor samples and a single punch of one control sample from an inflamed stromal area of the uterus. Among the samples were biopsies from one patient at three time points: at diagnosis, post treatment resection, and metastatic recurrence.

### 4.2. Immunohistochemistry (IHC)

Immunohistochemical staining of TMA sections was performed while using a BOND-MAX autostainer (Leica Microsystems, Wetzlar, Germany). Antibodies used for IHC were against NF-κB/p65RelA (1:500, Abcam, Cambridge, MA, USA), CD4 (1:500, Abcam), CD8 (pre-dilute, Leica), and PD-1 (1:500, Abcam). Two trained personnel manually scored TMA and whole sections. Percentage of positive cells was calculated per field. An IHC score was assigned to each sample by assigning a score according to the quartile metrics of a 0–3 scale and by averaging the scores of double punch samples.

### 4.3. DNA and RNA Sequencing of Patient Samples

FFPE tumor blocks from eight patients were sequentially sectioned at 10 µm and then mounted onto 21 slides by TACMASR. Two groups of five to ten adjacent slides from each tumor were used to macro-dissect tumor tissue and extract DNA or RNA. DNA and RNA extractions were performed while using the Qiagen GeneRead FFPE kit and Qiagen RNeasy FFPE kit (Venlo, Netherlands). Average DNA and RNA concentrations were 23.67 ± 22.78 ng/μL and 52.66 ± 50.22 ng/μL, respectively, with all samples passing standard nucleic acid quality metrics. DNA samples were prepared for next generation exome sequencing while using a Kapa Hyper into Agilent SureSelect XT protocol (Santa Clara, CA, USA). A custom Agilent SureSelect exome (55 Mb) was used to cover all coding regions of the genome in addition to tiling across regions that are commonly rearranged in cancer. The samples were sequenced on a HiSeq4000 (Illumina, San Diego, CA, USA) to an average depth of 220× coverage. RNA samples were prepared for next generation sequencing while using the Agilent SureSelectXT RNA Direct protocol, a capture-based approach that utilizes the Agilent SureSelect Exome (v6) + UTR (67 Mb) baits. The libraries were sequenced on a single lane on the HiSeq4000. Universal breast control RNA was purchased from Applied Biosystems (Waltham, MA, USA).

### 4.4. Sequencing Data Analysis

Primary analysis was performed while using the Translational Genomics Research Institute (TGen) Jetstream pipeline [44]. BCL files were converted to FASTQs with BCLConverter (Illumina, San Diego, CA, USA). DNA FASTQs were aligned to the human reference genome (build 37) with BWA-MEM (bwa v0.7.8) [45]. Base recalibration was performed with GATK v3.1-1 [46]. Duplicates were marked while using Picard v1.111 (http://broadinstitute.github.io/picard/) to generate BAMs, on which joint indel realignment was performed using GATK v3.1-1 [46] BAMs were used for the identification of somatic mutations (point mutations, insertions, deletions), structural variants (SVs), and copy number variants (CNVs). Variant calling was performed using Seurat (quality score > 30), Strelka, and MuTect, and annotated using GENCODE version 3 (Ensembl) and build 37.1 [47,48,49]. Final somatic SNVs were called by at least 2/3 callers. CNVs were predicted with tCoNuT (https://github.com/tgen/tCoNuT) with default thresholds: 0.58 for amplification and –0.99 for deletion. RNA FASTQs were aligned while using STAR 2.3.1z [50]. Longitudinal analysis was performed with LumosVar 2.0 that utilizes allele ratios and copy number states to predict somatic SNVs, INDels, and allelic copy number jointly across multiple samples from the same patient [51].

For hierarchical clustering analyses, matrix GCT files were generated while using HTSeqCounts values per sample from STAR aligner’s RNA-seq gene output and differentially expressed gene lists from DESeq2 (Bioconductor, Buffalo, NY, USA). The differentially expressed genes were obtained by comparing the tumor samples to a commercially available Universal breast control of a 55-year old Caucasian female (Applied Biosystems) using DESeq2, followed by filtering the data with a PASS flag based on Log2FC thresholds. Genes with Log2FC values ≤ −2 were considered to be downregulated and genes with Log2FC values ≥ 2 upregulated. Unsupervised clustering was performed with Spearman’s rank correlation and pairwise average-linkage while using gene- and sample-level normalization in GenePattern’s Hierarchical Clustering version 7.17 (Broad Institute, Cambridge, MA, USA). 

The Cancer Genome Atlas (TCGA) breast cancer dataset containing gene expression profiles of 1212 primary breast tumors samples with corresponding clinical annotations for 1090 samples were retrieved from NIH GDC Data Portal in order to conduct non-parametric survival analysis (https://portal.gdc.cancer.gov/projects/TCGA-BRCA). The RSEM-estimated gene expression levels in this dataset were used to generate expression scores for upregulated and downregulated genes, respectively, from the DESeq2 analysis. First, the RSEM values were converted to log2 and the log2 values were added for all of the genes per sample. The median of all the sums was used as a threshold. Samples with a sum ≥ median were deemed as ‘high’ and samples with a sum < median as ‘low’.

### 4.5. Statistical Analysis

The relationships between tumor pathologic features and molecular markers or between independent molecular markers were analyzed while using Fisher’s Exact test. An unpaired two-tailed student’s *t*-test was used to determine statistical significance. Kaplan-Meier survival plots were generated using GraphPad Prism (GraphPad Software, San Diego, CA, USA). The log-rank test was used to calculate the statistical significance between the survival curves. *p* values < 0.05 were considered to be statistically significant.

## 5. Conclusions

We identified PAGES-HBC in a small cohort of post-trastuzumab samples, correlating with patient outcomes in the TCGA HER2 breast cancer dataset. These results suggested that PAGES-HBC might be predictive of therapy response related patient outcome, but further work is needed in a larger cohort of patients to test the prognostic utility of PAGES-HBC. We conclude that PD-1 expression in post-trastuzumab tumors provides a measure of therapy response and that PAGES-HBC could be further developed as a biomarker for PD-1 positive therapy response in HER2+ breast cancer.

## Figures and Tables

**Figure 1 cancers-11-01566-f001:**
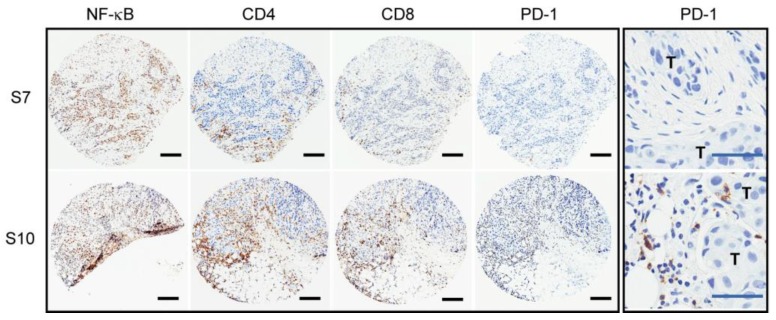
PD-1 positive immune environment in a subset of trastuzumab-treated HER2+ tumors. IHC staining images of tumors, S7 (PD-1 negative) and S10 (PD-1 positive) for NF-κB, CD4, CD8, and PD-1; black scale bar denotes 200 μm in length. The last panel contains higher magnification images of the PD-1 staining showing that PD-1-positive tumor immune cells in close proximity to tumor cells (large nuclei) in S10. T, tumor; blue scale bar denotes 50 μm in length.

**Figure 2 cancers-11-01566-f002:**
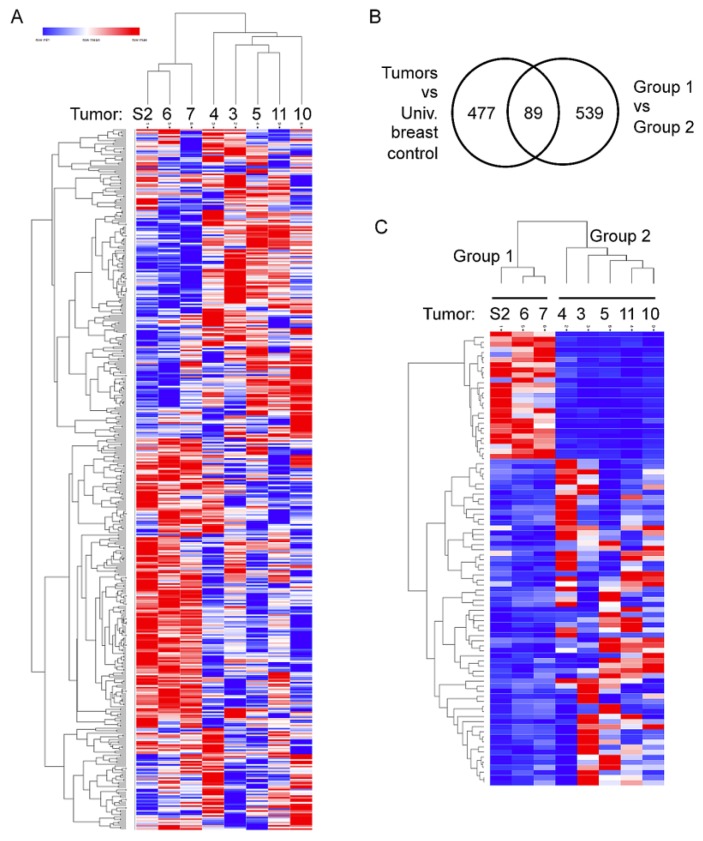
Two distinct groups of tumors based on differential expression clustering of the RNAseq data. (**A**), Heat map of unsupervised clustering; the first node in a hierarchical dendrogram separates S2, S6, and S7 tumors (Group 1) from S3, S4, S5, S10, and S11 tumors (Group 2). (**B**) Venn diagram showing 89 genes commonly found among the differentially expressed gene lists comparing between tumors vs. universal breast control and Group 1 vs. Group 2 tumors. (**C**) Heat map of supervised clustering of tumor using the 89 genes (genes listed in Appendix A).

**Figure 3 cancers-11-01566-f003:**
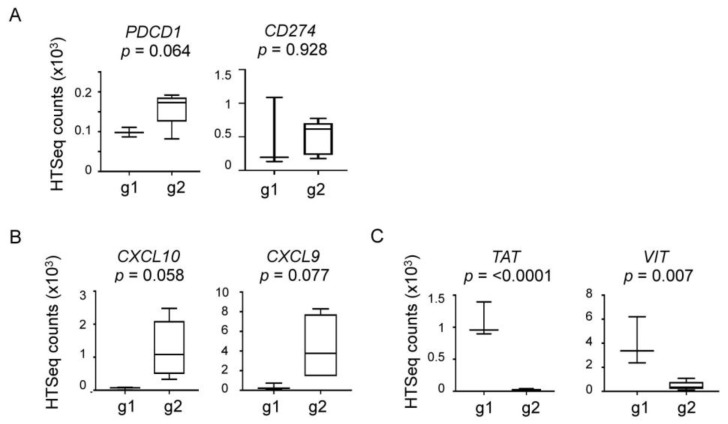
Expression levels of the genes in Group 1 vs. Group 2 tumors. (**A**) *PDCD1* and *CD274*, (**B**), *CXCL10* and *CXCL9*, and (**C**), *TAT* and *VIT*. Box plots were generated using the HTSeq counts of each gene. g1, Group 1; g2, Group 2; *p* values were determined using unpaired *t*-test.

**Figure 4 cancers-11-01566-f004:**
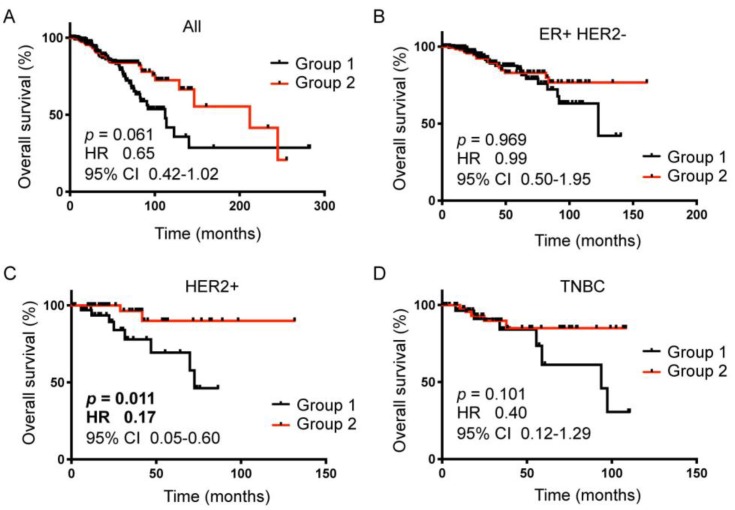
Kaplan-Meier survival analyses of the TCGA breast cancer patient data using PAGES-HBC Group 1 vs Group 2 compositions. (**A**), All breast cancer patients, (**B**), ER+ HER2− breast cancer patients, (**C**), HER2+ breast cancer patients, (**D**), Triple-negative breast cancer patients. Black line, Group 1 patients; red line, Group 2 patients; Kaplan-Meier analysis was done using Prism Graphad software; *p* values and hazard ratios were determined using Log-rank (Mantel-Cox) test.

**Figure 5 cancers-11-01566-f005:**
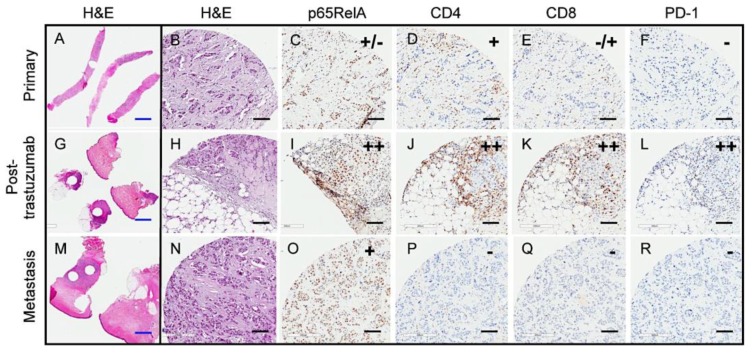
Transient immune response after trastuzumab therapy in longitudinal patient samples. (**A**–**F)**, primary needle biopsy, (**G**–**L**), post-neoadjuvant trastuzumab resected tumor, (**M**–**R**), recurrent tumor chest wall biopsy; **A**,**B**,**G**,**H**,**M**,**N**, **H** & **E**; **C**,**I**,**O**, NF-κB/p65RelA; **D**,**J**,**P**, CD4; **E**,**K**,**Q**, CD8; **F**,**L**,**R**, PD-1; blue scale bar denotes 2 mm in length, black scale bar denotes 100 μm in length.

**Figure 6 cancers-11-01566-f006:**
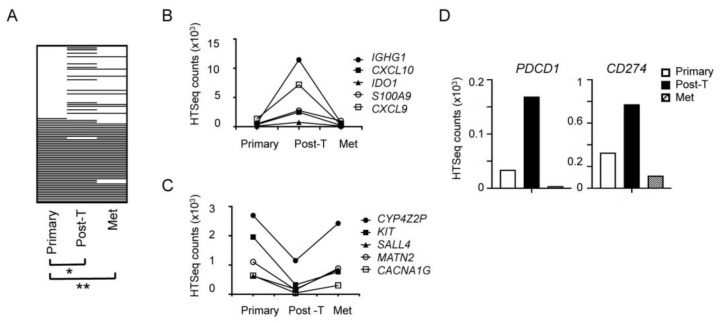
Dynamic PAGES-HBC changes in longitudinal samples of primary, post-neoadjuvant trastuzumab treated, and recurrent tumors from a single patient. (**A**), Heat map of PAGES-HBC, white bar, Group 1 level expression; black bar, Group 2 level expression; * *p* = 0.03; ** *p* = 0.65. (**B**), Expression of the genes that are transiently upregulated in trastuzumab-treated sample, (**C**), Expression of the genes that are transiently downregulated in trastuzumab-treated sample. (**D**), Transient upregulation of *PDCD1* (PD-1) and *CD274* (PD-L1) in trastuzumab-treated sample. Post-T, post-trastuzumab; Met, metastasis.

**Figure 7 cancers-11-01566-f007:**
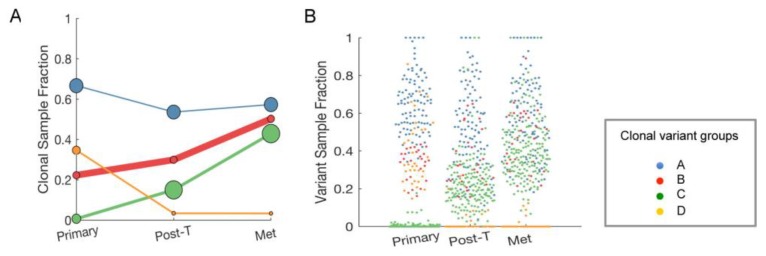
Evolution of single nucleotide variants (SNVs) in longitudinal samples. (**A**), The proportion of cells expected to have the SNVs in each clonal variant group in each sample. LumosVar2 was used to assign mutations to clonal variant groups. Clonal variant group A (blue) was present in all of the tumor cells; clonal variant groups B (red) and C (green) increased in abundance overtime; clonal variant group D (orange) was present in the primary tumor but is mostly not detectable at the later time-points. The size of the circles is proportional the number of mutations detected in the sample in that clonal variant group, and the thickness of the line represents the proportion of copy number alterations assigned to the clonal variant group. (**B**), The proportion of cells in the sample estimated to have the mutation is plotted for each sample. Each point represents a mutation, and the color indicating the clonal variant group. Post-T, post-trastuzumab; Met, metastasis.

**Table 1 cancers-11-01566-t001:** Cohort clinical characteristics.

Tumor ID	Age	Tumor Size (cm)	Tumor Grade	Lymph Node (+/total)	HER2 Scores	ER (%)	Neoadjuvant Trastuzumab (Duration)
S2	58	2.2	III	5/19	3+	40	Y (3 months)
S3	55	1.5	II	2/14	3+	90	Y (4 months)
S4	38	1.0	III	0/13	3+	0	Y (2 months)
S5	61	1.5	III	0/0	3+	80	Y (3 weeks)
S6	48	0.2	NP	0/8	3+	0	Y (1 year)
S7	59	NP	III	5/21	3+	0	Y (3 months)
S10	36	NP	NP	NP	3+	0	Y (3 months)
S11	43	4.5	III	1/6	3+	0	Y (NP)

NP, Information not provided.

**Table 2 cancers-11-01566-t002:** Summary of Immunohistochemistry (IHC) markers and putative driver mutations.

Tumor ID	NF-κB p65	CD4	CD8	PD-1	# SNV/CNV	Putative Driver Mutations
S2	++	–	++	–	8	N/A
S3	++	++	++	–	20	HER2 copy gain; PTEN HD
S4	++	+	++	+	8	HER2 overexpression
S5	++	++	++	+	16	HER2 copy gain
S6	ND	ND	ND	–	3	N/A
S7	++	+	+	–	7	PIK3CA aSNV
S10	++	++	++	++	17	HER2 copy gain/SV; TP53 iSNV
S11	++	++	++	+	25	PIK3CA aSNV, TP53 iSNV

N/A, Not apparent; ND, Not determined; HD, homozygous deletion; aSNV, activating single nucleotide variant; SV, single copy variant; iSNV, inactivating single nucleotide variant. Complete SNVs/CNVs are listed in Appendix A.

**Table 3 cancers-11-01566-t003:** PD-1-Associated Gene Expression Signature in HER2+ Breast Cancer (PAGES-HBC) Group 1 and Group 2 patients in the The Cancer Gene Atlas (TCGA) breast cancer data set.

BC Subtype	*n*	Group 1	Group 2
All	1090	250 (23%)	309 (28%)
ER+HER2−	590	131 (22%)	159 (27%)
HER2+	181	36 (20%)	53 (29%)
TNBC	160	32 (20%)	51 (32%)

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
