# Peer review of "PD-1-Associated Gene Expression Signature of Neoadjuvant Trastuzumab-Treated Tumors Correlates with Patient Survival in HER2-Positive Breast Cancer"

_cancers, 2019, doi:10.3390/cancers11101566_

Round 1

Reviewer 1 Report

The proposed research study attempted to identify predictive biomarkers for trastuzumab-elicited tumor immune responses by first performing genomic and immunohistochemical profiling (for nuclear NF-κB/p65RelA, CD4, and CD8 T cell markers) of tumors from 8 patients with HER2+ diseases after completion of neoadjuvant trastuzumab. DNA sequencing was attempted to identity driver mutations. Differential expression of 89 genes (PAGES-HBC) was able to stratify HER2+ patients in two groups and was validated with TCGA dataset. PAGES-HBC was not statistically different from post-trastuzumab treated tumors when compared with biopsies. PAGES-HBC may serve as a prognostic predictor of trastuzumab response in HER2+ breast cancer patients who may also potentially benefit from added anti-PD-1 therapy. The research designs and rationales are scientific sound.

It would be helpful to know the clinical TNM staging of recruited patients. Are patients with trastuzumab-resistant tumors (partial response or stable disease) also with higher PD-1 levels?

Two genes (TAT and VIT) were found to be differentially expressed. Is there additional information regarding the function and potential role of these genes?

Reviewer 2 Report

The study from William et al attempted to identify predictive biomarkers for trastuzumab-elicited tumor immune responses by genomic and immunohistochemical profiling of tumors from 8 patients who have completed multiple rounds of neoadjuvant trastuzumab. The authors found that all tumors had an activated tumor  immune microenvironment  which was reflected by the positive staining of NF-κB/p65RelA, CD4, and CD8 T. However, it was found that only 4 out of 8 tumors were positive for the PD-1. In addition, exome sequencing of those samples showed no specific driver mutations  correlating with PD-1 positivity but RNA sequencing data revealed two distinct groups, of which Group 2 represented the PD-1 positive tumors. A gene expression signature derived from this clustering composed of genes stratified HER2+ breast cancer patients in the TCGA dataset (PAGES-HBC). Patients with the Group 2 PAGES-HBC profile had a more  favorable survival. Analysis of three longitudinal samples from a single patient showed that PAGES-HBC  may be transiently induced by trastuzumab. The authors concluded that PAGES-HBC could be further developed as a prognostic predictor of trastuzumab response in HER2+ breast cancer patients who may also potentially benefit from anti-PD-1 associated immunotherapy.

1) Figure 1. The IHC result showing that S10 sample is positive for PD-1 staining is not convincing. It is an overstament that PAGES-HBC could be potentially benefit from added anti-PD-1 therapy. No data supports the claim.

2) The molecular mechanisms by which trastuzumab-elicited tumor immune responses , especially PD-1 expression, was not discussed. If the 89-gene signature was not specific to HER2+ breast cancer or to any other subtype, 89 gene signature expression should not be specific to HER2 treatment.

3) It is not clear how three longitudinal samples from a single patient were collected. Given intra-tumor heterogeneity in tumors, multiple samples from the different area of the same tumor should be collected.

4)The major concern is that several key conclusion were based on the results from a single patient, such as Fig.5-7. The integrated analysis of the transient immune response, dynamic PAGES-HBC changes and evolution of single nucleotide variants of three longitudinal samples was from a single patient. The conclusion is far-fetched that trastuzumab-elicited tumor immune activation may be transient.

Reviewer 3 Report

This study aimed to identify the predictive biomarkers for trastuzumab-elicited tumor immune response and prognosis. They assayed the gene expression profiles in several tumors and identified 89 genes as markers to classify patients in two different groups (group 1 and 2). Their results showed that group 2 tumors express higher immune and inflammatory genes, and is correlated with better prognosis of HER2+ breast cancer patients. After validating by TCGA datasets, they further performed SNV analysis and proposed a concept of clonal expansion. This is an interesting study providing novel evidence with clinical significance.   

In Figure 5, it will be easier for readers if the authors could label the source of tissue directly along with all the figure blocks.

The authors showed SNVs in longitudinal samples in Figure 7. They may consider to modify the group label in alphabetical, since group 1 and 2 have been used in presenting tumors with different gene expression profiles and may be confused by readers.

The authors may consider to perform an integrated analysis to demonstrate the correlation between gene expression signatures and SNVs.

How many genes in this study are overlapped with the genes used in currently-used clinical test.

Round 2

Reviewer 2 Report

The revised manuscript has been approved.

Author Response

Thank you for approving our revised manuscript.